# Analysis of Factors Associated with Subjective Mild Cognitive Impairment (MCI) among Older Adults Resident in the Community

**DOI:** 10.3390/ijerph191610387

**Published:** 2022-08-20

**Authors:** Eun Jeong Hwang

**Affiliations:** Department of Nursing, Sehan University, 1113 Samho-eup, Yeongam-gun 58447, Korea; ejhwang@sehan.ac.kr; Tel.: +82-10-5305-1581; Fax: +82-61-469-1317

**Keywords:** mild cognitive impairment, older adult, personal characteristics, health promotion activities, quality of life

## Abstract

This study explored the associated factors of mild cognitive impairment (MCI) in older adults, focusing on personal characteristics and health promotion activities. The research design of this study was a non-experimental, cross-sectional design. This study used secondary data from the 2019 community health survey conducted by the Korea Disease Control and Prevention Agency (KDCA). In this study, 20,041 older adults with subjective MCI and 52,587 healthy older adults—a total of 72,628 subjects—were analyzed as the final subjects in this study. The data were analyzed by using descriptive statistics, a chi-square test, an independent *t*-test, and logistic regression. The results indicate that the probability of experiencing subjective MCI significantly decreased with an increase in monthly income (odds ratio (OR) = 1.71, 95% confidence interval (CI) = 1.19–2.46); increased with an increase in depressive symptoms (odds ratio (OR) = 1.09, 95% confidence interval (CI) = 1.04–1.15); and decreased with an increase in the quality of life level (OR = 0.90, 95% CI = 0.82–0.99). Through the results of this study, several factors related to MCI in older adults were identified. If these related factors are properly managed, the possibility of MCI in older adults may be reduced. Therefore, MCI in older adults should be addressed as a preventable disease rather than a natural aging process.

## 1. Introduction

In Korea, the number of dementia patients continues to increase due to an aging population, and in 2019, the average prevalence of dementia among older adults over 65 in Korea was 10.3% [1]. Currently, mild cognitive impairment (MCI) is considered to be important in relation to the state of cognitive function between normal aging and dementia [2]. MCI represents an initial stage in the continuum of cognitive aging, where the person can recognize that he or she has a problem with cognitive function [2]; this accounts for about 22.7% of older adults over 65 in Korea [1]. Older people may normally experience a decrease in cognitive function due to aging, but if this condition is left untreated, MCI may occur; those with MCI are included in the high-risk group for dementia, with a very high probability of progressing to dementia [3]. Eleven to thirty-three percent of patients with MCI develop dementia [4]. In particular, according to the study of Petersen [2], it has been reported that MCI may occur in 15% to 20% of older adults over 65 years of age; each year, 8% to 15% of patients with MCI develop dementia. In order to effectively prevent dementia, it is necessary to assess and manage characteristics in a timely manner at the initial stage of cognitive decline in the continuum of cognitive aging. Early cognitive decline can be somewhat difficult for others to notice. Therefore, we should consider the subject if they feel ‘different compared to the past’ [4]. Accordingly, in the cognitive function evaluation, the importance of how the subject feels and evaluates their own cognitive function change is ‘subjectively’ increasing [4]. A term with the key word or meaning of “subjective” has been proposed for implementation in order to define the difference between cognitive decline due to normal aging and that which is due to MCI [4]. The early detection of MCI and the appropriate intervention are very important as they can slow the progression of dementia or improve the symptoms [5].

Previous studies have suggested various risk factors for MCI. In a nationwide survey of older adults over 60, Jia et al. [6] reported that MCI had similar risk factors to those of dementia, such as old age, sex, family history, rural residence, low education, living alone, smoking, and chronic diseases. Byeon [7] suggested a number of major risk factors for MCI, including age, gender, educational background, subjective health, marital status, income level, smoking, and regular exercise, using the restricted Boltzmann machine (RBM) artificial neural network-based prediction model. Wong et al. [8] suggested low education, living alone, smoking, and a high-fat diet as risk factors for MCI in relation to lifestyle. Alegret et al. [9] reported that emotional characteristics such as depression and anxiety were highly correlated with subjective cognitive decline. According to the study results of Song et al. [10], it was reported that physical exercise had a significant effect on MCI. Reviewing a number of previous studies makes it possible to classify the risk factors for MCI into personal characteristics and health promotion activity characteristics. This study explored the associated factors of subjective MCI in older adults and focused on personal characteristics and health promotion activities.

## 2. Materials and Methods

### 2.1. Design

The study was conducted by using a non-experimental, cross-sectional research design.

### 2.2. Data Collection and Procedures

The researcher extracted some of the data matching this research purpose from secondary data, collected through the 2019 community health survey conducted by the Korea Disease Control and Prevention Agency (KDCA). This survey tool was developed through research and analysis with the participation of experts, professors, and administrators in various fields. This survey has been conducted annually since 2008; the survey data were collected using around 250 public health centers networking across the country. The target population of this survey comprised individuals aged 19 years and older. As for the survey sampling method, an average of 900 people per public health center was selected using a multi-stage cluster sampling method. The survey data collection process was conducted through a face-to-face interview with trained interviewers visiting each selected household, using a computer-assisted personal interviewing technique. The interviewers explained the purpose of the survey and the confidentiality measures used in the survey to the participants. Subsequently, they collected data through one-on-one, face-to-face interviews with the participants. The data in this survey were collected using self-report structured questionnaires. The survey period was from 6 August to 31 October 2019.

### 2.3. Selection Criteria for Subjects with Subjective MCI

Petersen et al. [11] suggest that the diagnosis of MCI be made if the patient meets the following criteria: (1) complaint of defective memory, (2) normal activities of daily living, (3) normal general cognitive function, (4) abnormal memory for their age, and (5) absence of dementia. In this study, the subjects with MCI were selected by applying Paterson’s diagnostic criteria of MCI.

### 2.4. Research Instrument

#### 2.4.1. Health Promotion Activities

Health promotion activities consisted of smoking type, period of smoking, frequency of drinking alcohol in the past year, physical activities, and physical activity days per week. Smoking types were divided into three groups: smokers, stop-smokers, and non-smokers. To classify these groups, the questionnaire asked whether or not smoking referred to current smoking; those who answered ‘everyday’ and ‘sometimes’ were classified as smokers; those who answered ‘smoked in the past but not currently’ were classified as stop-smokers; those who answered ‘not applicable’ were classified as non-smokers. Among the stop-smokers group, smoking cessation factors of less than one year were excluded because the smoking cessation period was more than one year. The smoking period of the subjects in this study was calculated and divided between smokers and stop-smokers. For smokers, the smoking period was the number of years of smoking, calculated by subtracting the age at which smoking started from the current age of the subjects in the study. The smoking period of non-smokers was converted into ‘years’ from the ‘months’ of the smoking period which they indicated. Regarding alcohol consumption in the past year, the responses to the question ‘have you drunk alcohol in the past year?’ were classified as non-drinking = 1; less than once a month = 2; once a month = 3; 2–4 times a month = 4; 2–3 times a week = 5; and 4 times or more a week = 6. As to the question of whether or not the respondents performed 10 min or more of moderate-intensity physical activities in a recent week, ‘non-performed’ was classified as 1 and ‘performed’ was classified as 2. The physical activity days per week referred to the number of days the respondents had performed moderate-intensity exercise for at least 10 min or more over a recent week.

#### 2.4.2. Body Mass Index

Body mass index (BMI) was calculated by dividing weight (kg) by height squared (m^2^) [12], using the subject’s height and weight values. According to the World Health Organization [12], a BMI of 18.5 or less is classified as underweight; 18.5 to 24.9 is normal; 25 to 29.9 is overweight and classed as pre-obese; 30 to 34.9 is moderately overweight and classed as obese I; 35 to 39.9 is severely overweight and classed as obese II; and over 40 is severely overweight and classed as obese III [12].

#### 2.4.3. Depression Symptoms

These were assessed using the following nine items: “no interest in or fun at work”; “sinking feeling, depression, and hopelessness”; “difficulty falling asleep or sleeping too much”; “feeling tired”; “lack of appetite or overeating”; “considering oneself worthless and a harbinger of misery”; “difficulty concentrating on newspapers or television”; “nervousness, anxiety, or too much wandering”; and “believing that death is preferable to living or experiencing thoughts about hurting oneself”. Each item was rated on a four-point Likert scale (1 = never, 2 = felt for several days, 3 = felt for over a week, and 4 = felt almost every day), with higher scores indicating higher levels of depression. Cronbach’s α of the depression instrument was 0.81 in the present study.

#### 2.4.4. Subjective Health Status

The subjective health status of older adults living alone was an observational variable assessed by one question about the participants’ health condition at the time of the survey. Participants indicated their state of health on a five-point Likert scale (1 = very bad, 2 = bad, 3 = neutral, 4 = good, and 5 = very good).

#### 2.4.5. Quality of Life (QoL)

The quality of life of older adults living alone was an observational variable assessed by one question as to participants’ QoL level at the time of the survey. Participants marked their happiness on a 10-point graphic rating scale (1 = terribly unsatisfied and 10 = extremely satisfied), with higher scores indicating higher QoL levels.

### 2.5. Ethical Considerations

The data used in this study are secondary data collected through the 2019 community health survey conducted by the Korea Disease Control and Prevention Agency (KDCA). The participants were provided with an explanation of the proposed study and informed that they were free to withdraw from the study at any time without prejudice. The identification data of the study subjects were kept separately at the health centers and not made available to those conducting the study. The Korea Disease Control and Prevention Agency (KDCA) evaluates the purpose of the study and whether it is appropriate and provides data without personal identifiable information free of charge to researchers in accordance with the official procedure. Before conducting this study, approval was obtained from the institutional review board of Sehan University (approval number SH-IRB 2021-102).

### 2.6. Statistical Analyses

The data were analyzed by using IBM SPSS version 21 software (IBM Corporation, Armonk, NY 10504, USA). An inferential statistical analysis was conducted using the chi-square test and the *t*-test. Logistic regression was performed to determine the associated factors of older adults with MCI.

## 3. Results

### 3.1. Sample Characteristics

The present sample comprised 72,628 subjects: 20,041 (27.6%) subjects with subjective MCI and 52,587 (72.4%) healthy subjects (Table 1). There were significant differences in the sociodemographic characteristics of the two groups (*p* < 0.001). Among the subjects with subjective MCI, 7437 (10.2%) were male and 12,604 (17.4%) were female. Among the healthy subjects, 22,979 (31.6%) were male and 29,608 (40.8%) were female. Regarding age group, the highest proportion was 75 to 79 years old for subjects with subjective MCI and 65 to 69 years old for healthy subjects, indicating significant differences in age between the two groups (*χ*^2^ = 816.54, *p* < 0.001). The average age was 75.53 ± 6.64 years for subjects with subjective MCI and 73.95 ± 6.42 years for healthy subjects, indicating significant differences in the average age between the two groups (*t* = −28.82, *p* < 0.001). Although the educational category of elementary school graduates had the highest percentage in both groups, there was a significant difference in the educational level between the two groups (*χ*^2^ = 579.42, *p* < 0.001). Regarding marital status, 11,836 (16.3%) subjects with subjective MCI and 34,654 (47.7%) healthy subjects were married, with these categories having the highest percentages in these two groups (*χ*^2^ = 323.94, *p* < 0.001). Although the employment status category of unemployment had the highest percentage in both groups, there was a significant difference in employment status between the two groups (*χ*^2^ = 178.45, *p* < 0.001). In both groups, the highest percentage was observed for those not eligible for basic livelihood rights (*χ*^2^ = 102.63, *p* < 0.001). Regarding monthly average income, the highest proportion was KRW 500,000 to 990,000 for subjects with subjective MCI, and KRW 1,000,000 to 1,990,000 for healthy subjects, indicating significant differences in the average monthly income between the two groups (*χ*^2^ = 293.20, *p* < 0.001).

### 3.2. Comparison of Variables between the Two Groups

Significant differences between the two groups were observed for health promotion activities, body mass index, depressive symptoms, and subjective health status (*p* < 0.001). There are presented in Table 2. In terms of health promotion activities, the two groups differed significantly in terms of smoking type (*p* < 0.001), period of smoking (*p* = 0.025), age started smoking (*p* < 0.001), frequency of drinking alcohol in the past year (*p* < 0.001), physical activities (*p* < 0.001), and physical activity days per week (*p* < 0.001). Regarding smoking type, the highest percentage was observed for the non-smokers in both groups (*χ*^2^ = 93.67, *p* < 0.001). The average smoking period of the subjects with subjective MCI was 14.91 (±19.80) years, which was significantly shorter than the average smoking period of the healthy subjects of 15.54 (±19.96) years (*t* = 2.24, *p* = 0.025). However, the age at which the subjects started smoking among those with subjective MCI was 22.74 (±7.73) years old, which was significantly earlier than the age of the healthy subjects, which was 22.30 (±6.96) years old (*t* = −4.11, *p* < 0.001). In relation to the frequency of drinking alcohol during the past year, the highest number in both groups was non-drinking (*χ*^2^ = 98.96, *p* < 0.001). Regarding physical activities, the highest percentage was observed for those who ‘performed’ in both groups (*χ*^2^ = 67.16, *p* < 0.001). The average days of physical activity per week among the subjects with subjective MCI was 5.13 (±4.37) days, which was significantly lower than the average days of physical activity per week among the healthy subjects, which was 5.52 (±4.42) days (*t* = 10.81, *p* < 0.001). The average BMI of the subjects with subjective MCI was 24.09 (±5.34) kg/m^2^, which was significantly lower than the BMI of the healthy subjects, which was 24.23 (±6.21) kg/m^2^ (*t* = 2.83, *p* < 0.001). The average of the depression symptoms of the subjects with subjective MCI was 12.60 (±3.96), which was significantly higher than that of the depression symptoms among the healthy subjects, which was 10.97 (±2.86) (*t* = −53.19, *p* < 0.001). Regarding subjective health status, the highest proportion was ‘bad’ for the subjects with MCI, and the highest proportion was ‘neutral’ for the healthy subjects, indicating significant differences in subjective health status between the two groups (*χ*^2^ = 1732.48, *p* < 0.001). The average quality of life for the subjects with subjective MCI was 6.31 (±1.87), which was significantly lower than the quality of life for the healthy subjects, which was 6.86 (±1.79) (*t* = 35.68, *p* < 0.001).

### 3.3. Logistic Regression Analyses

The results for the logistic regression of the general characteristics, health promotion activities, BMI, depressive symptoms, subjective health status, and quality of life for the subjects with subjective MCI and the healthy subjects are presented in Table 3. The model was constructed with MCI and healthy as dependent variables; general characteristics, health promotion activities, BMI, depressive symptoms, subjective health status, and quality of life were independent variables. The model (−2 Log L = 974.32, chi-square = 69.85, *p* < 0.001) met the convergence criterion for logistic regression. The significant predictive factors influencing subjective MCI in the subjects were as follows. The probability of experiencing subjective MCI was 1.7 times higher for subjects with a monthly income of KRW 1,000,000~1,990,000 compared to subjects with a monthly income of KRW 3,000,000 or more (OR = 1.71, 95% CI = 1.19–2.46). Furthermore, the probability of experiencing MCI increased with an increase in depressive symptoms (OR = 1.09, 95% CI = 1.04–1.15) and decreased with an increase in the quality of life level (OR = 0.90, 95% CI = 0.82–0.99).

## 4. Discussion

The results of this study indicate that the average age of the older adults with subjective MCI was significantly higher than that of the healthy older adults (*p* < 0.001). In several related studies [6,8,13], age was an important risk factor for MCI and dementia. If cognitive impairment symptoms are recognized by the subject or those around them, the pathological changes in the brain would have already been progressing for decades [4]. Therefore, cognitive impairment can be more precisely recognized over time and should be closely related to age. The results of this study showed that monthly income was one of the significant associated factors of MCI. This result is consistent with Van Leeuwen et al. [14], who found that financial resources influence older people’s quality of life, independence, and access to a comfortable life. It is also consistent with Kim’s study [15], which found that the household income of older adults affects quality of life. Hwang and Sim [16] also stated that income had a significant effect on the happy life of older adults. It would be very unfortunate if older adults had to work despite their old age because of poverty. Dingemans and Henkens [17] found that older people from poor socioeconomic backgrounds were forced to work under adverse conditions because of limited career options. Therefore, income can have a profound effect on the physical and mental health of older adults.

As a result of this study, it was found that the subjects with subjective MCI had a significantly shorter average smoking period than healthy subjects (*p* = 0.025). Hong [18] supported the results of this study by providing evidence that the greater the number of days of illness, the higher the probability of success in quitting smoking. Moreover, Song and Lee [19] stated that the higher the subjective health level of smokers, the lower the probability of success in quitting smoking. In Ahn’s [20] study, it was reported that the intention to quit smoking was significantly lower in elderly subjects than in younger subjects. Taken together, it seems that even if you get old, you do not want to quit smoking if you do not get sick, but you have no choice but to quit when you get sick. Therefore, it is considered that the reason for the short smoking period among the subjects with MCI in this study was that they unwillingly quit smoking due to health problems. However, several previous studies [6,7,8] reported that smoking has a profound effect on MCI. Kumboyono et al. [21] stated that smoking increased the premature associated senescence phenotype of circulating endothelial progenitor cells (EPCs), which may contribute to the diminished bioavailability of the mature EPCs of the smoker, thereby reducing the potency of vascular maintenance and repair. Smoking lowers the oxygen concentration in the blood, leading to hypoxia [22], weakening blood vessel function [21], and worsening bodily functions. In addition, it can be confirmed that smoking is closely related to many predictors of MCI in older adults, such as psychologically induced depression [23]. The results of this study also showed that the age that the subjects started smoking was significantly earlier among those with MCI than those who were healthy (*p* < 0.001), which was similar to the results of previous studies.

As a result of this study, it was found that the subjects with subjective MCI had significantly shorter weekly physical activity days than the healthy subjects (*p* < 0.001). According to Nuzum, Corona, Zeller, Melrose, and Wilkins [24], physical activity in older adults is effective in improving their overall health and cognitive function, independent functioning, and psychological health, thereby supporting the results of this study. In addition, they emphasized the need for physical activity intervention in subjects experiencing cognitive decline. Erickson [25] reported that as a result of providing exercise intervention to 120 older adults for one year, aerobic exercise increased hippocampal capacity and confirmed significant changes in memory function and serum brain-derived neurotrophic factors. Makizako [26] reported the possibility of an increase in blood flow in the frontal lobe from exercise and an increase in blood flow in the motor area according to hand and foot movements. Doi et al. [27] stated that cognitive leisure activity had a positive effect on cognition in MCI. It should be noted that Choi and Youn [28] stated that physical activity among older adults should not be too strenuous and should involve soft exercises of a low intensity that are within the range of joint motion; this is helpful in improving the patient’s cognitive function, such as recall ability. Several studies [29,30] have reported that lifestyle interventions, such as moderate physical exercise, cognitive training, and diet, have induced a symptomatic benefit in MCI. Through these studies, it was confirmed that various physical activities had a positive effect on cognitive impairment, and the results of this study were also consistent.

The results of this study showed that BMI was a significant associated factor of MCI. The results of this study were consistent with the results of a study by Joo et al. [31], who stated that being underweight could be an important indicator in identifying the possibility of progressing from MCI to Alzheimer’s disease. Kadey et al. [32] found that decreased BMI over five years was strongly associated with dementia or MCI. Therefore, they insisted on interventions and behaviors to increase BMI to improve long-term cognitive health in older adults. In related studies [13,31], lower BMI was identified as a risk factor for MCI and Alzheimer’s disease. According to Joo et al. [31], a lower BMI was found to act as more of a risk factor for MCI than a normal BMI, especially in older adults, females, and hypertension patients. On the other hand, Yuan et al. [13] stated that men with a high BMI were found to have an increased risk of MCI. Moreover, Sanderlin, Todem, and Bozoki [33] stated that obesity and comorbidities were associated with certain neuropsychiatric symptoms, including MCI. Therefore, in order to prevent MCI, it is important to maintain a BMI in a normal range that is neither high nor low. The results of this study show that symptoms of depression were significant associated factors of MCI. The results of this study were also consistent with the results of a study by Ma [34], who stated that MCI patients have a high prevalence of depression and that patients with MCI and concomitant depression have more pronounced cognitive deficits and progress more often to dementia than MCI patients without depression. Furthermore, Lang et al. [35] reported that higher depression was seen in amnestic MCI compared to the cognitively normal group for both ethnicities, after controlling for age, education, gender, and the Mini-Mental State Examination score. Through related studies, it was confirmed that quality of life had a close effect on MCI. In particular, the quality of life during old age is important as it is closely related to health, cognitive impairment, and dementia. The quality of life of older adults was found to be significantly related to present health status and depressive symptoms [16]. Anderson [29] stated that the symptoms of MCI comprise a lower quality of life, higher depressive symptoms, social avoidance strategies, and withdrawal from social participation. According to Gong and Tao [36], biopsychosocial holistic care intervention improved the cognitive function and quality of life of older adults with MCI.

Through a review of this study and related studies, it was confirmed that multi-faceted interventions are needed to prevent MCI in older adults. MCI should not be regarded as a natural process of aging and should not depend on the individual or their family’s efforts alone. Currently, older adults in poverty pose a serious problem that can harm not only economic status, but also physical and mental health. There should be interventions to solve the poverty of older adults. Research, various strategies, and policies to prevent MCI in older adults should be continually developed. At the same time, social and policy efforts should be made to improve the overall quality of life of the elderly due to the increase in the elderly population worldwide.

### 4.1. Limitations

This study had some limitations. The data used in this study are non-experimental cross-sectional data; so, it is somewhat difficult to infer a causal relationship. The data in this study were collected from public health centers across the country, and the same structured questionnaire was used, but there may be exogenous variable interventions due to the influence of subject characteristics and environmental factors. In addition, the possibility of response bias cannot be excluded because the data were imported in a self-reported format. As secondary data originally collected for other purposes were used, attempts to generalize the results of this study should be made cautiously.

### 4.2. Implications for Further Research

Repetition studies using research tools developed for subjects with MCI are suggested for further research. Moreover, an experimental study to verify the effectiveness of the MCI improvement program, applying the useful factors shown in this study, is proposed for further research.

## 5. Conclusions

This study explored the associated factors of subjective MCI in older adults, focusing on personal characteristics and health promotion activities. The findings of this study show that MCI in older adults can be prevented if various economic, physical, and social strategies to enhance a healthy life are applied. MCI in older adults should be addressed as a preventable disease rather than a natural aging process. In-depth research and intervention strategies for the overall quality of life should be continuously developed in order for older adults to live the rest of their lives in a healthy way. Relevant academia experts and policy makers should try to address the problem of cognitive impairment among older adults.

## Figures and Tables

**Table 1 ijerph-19-10387-t001:** Comparison of characteristics of older adults with subjective MCI and healthy older adults (*n* = 72,628) ^1^.

Characteristics	Older Adults with Subjective MCI ^2^ (*n* = 20,041)		Healthy Older Adults (*n* = 52,587)		*χ*^2^ or *t*	*p*
	*n*	%	*n*	%		
Gender					258.76	<0.001
Male	7437	10.2	22,979	31.6		
Female	12,604	17.4	29,608	40.8		
Age group (years)					816.54	<0.001
65–69	4497	6.2	15,927	21.9		
70–74	4644	6.4	13,676	18.8		
75–79	5161	7.1	12,069	16.6		
80–84	3780	5.2	7449	10.3		
85–89	1569	2.2	2748	3.8		
≥90	390	0.5	718	1.0		
M ± SD	75.53 ± 6.64		73.95 ± 6.42		−28.82	<0.001
Range	65–103		65–105			
Educational level					579.42	<0.001
Illiteracy	4475	6.2	8622	11.9		
Elementary school	8638	11.9	21,455	29.6		
Middle school	3134	4.3	9356	12.9		
High school	2618	3.6	8777	12.1		
College or higher	1152	1.6	4317	5.9		
Marital status					323.94	<0.001
Married	11,836	16.3	34,654	47.7		
Divorced	707	1.0	1816	2.5		
Widowed	7404	10.2	15,812	21.8		
Never married	83	0.1	271	0.4		
Employment status					178.45	<0.001
Employed	7266	10.0	21,920	30.2		
Unemployed	12,769	17.6	30,640	42.2		
Eligibility for basic livelihood right					102.63	<0.001
Yes	1293	1.8	2575	3.5		
In the past	251	0.3	426	0.6		
No	18,480	25.5	49,549	68.3		
Monthly income (KRW 10,000 ^3^)					293.20	<0.001
<50	1780	2.9	3699	5.9		
50–99	5633	9.0	12,377	19.8		
100–199	4878	7.8	12,897	20.7		
200–299	2204	3.5	6898	11.1		
≥300	2945	4.7	9095	14.6		

^1^ Missing data were excluded. ^2^ MCI = mild cognitive impairment. ^3^ KRW 1200 = USD 1.

**Table 2 ijerph-19-10387-t002:** Comparison of health promotion activities, BMI, depression symptoms, subjective health status, and quality of life between older adults with subjective MCI and healthy older adults (*n* = 72,628) ^1^.

Variables		Categories	Older Adults with Subjective MCI ^2^ (*n* = 20,041)		Healthy Older Adults (*n* = 52,587)		χ^2^ or *t*	*p*
			*n*	%	*n*	%		
Health promotion activities	Smoking type	Smokers	1534	2.1	4867	6.7	93.67	<0.001
		Stop-smokers	5065	7.0	14,350	19.8		
		Non-smokers	13,442	18.5	33,370	45.9		
	Period of smoking (years)		14.91 ± 19.80		15.54 ± 19.96		2.24	0.025
	Age started smoking		22.74 ± 7.73		22.30 ± 6.96		−4.11	<0.001
	Frequency of drinking alcohol (for the past year)	Non-drinking	12,331	17.0	30,889	42.5	98.96	<0.001
		Less than once a month	2578	3.5	6329	8.7		
		Once a month	1055	1.5	2958	4.1		
		2–4 times a month	1579	2.2	4728	6.5		
		2–3 times a week	1186	1.6	3603	5.0		
		≥4 times a week	1310	1.8	4073	5.6		
	Physical activities	Non-performed	4685	6.5	10,827	14.9	67.16	<0.001
		Performed	15,355	21.1	41,757	57.5		
	Physical activity days per week		5.13 ± 4.37		5.52 ± 4.42		10.81	<0.001
Body mass index			24.09 ± 5.34		24.23 ± 6.21		2.83	<0.001
Depression symptoms			12.60 ± 3.96		10.97 ± 2.86		−53.19	<0.001
Subjective health status		Very bad	2584	3.56	4049	5.58	1732.48	<0.001
		Bad	7844	10.80	14,865	20.47		
		Neutral	6815	9.38	21,685	29.86		
		Good	2572	3.54	10,838	14.92		
		Very good	224	0.31	1146	1.58		
Quality of life			6.31 ± 1.87		6.86 ± 1.79		35.68	<0.001

^1^ Missing data were excluded. ^2^ MCI = mild cognitive impairment.

**Table 3 ijerph-19-10387-t003:** Logistic regression model for associated factors of subjective MCI.

Variables	OR	95% CI
Gender (male/female)	0.94	0.56–1.58
Age	1.02	0.99–1.05
Education level		
Illiteracy	0.82	0.49–1.36
Elementary school	0.99	0.65–1.52
Middle school	0.85	0.60–1.21
High school	0.93	0.66–1.31
College or higher	Referent	
Marital status		
Married	1.03	0.20–5.25
Divorced	1.14	0.28–4.69
Widowed	0.41	0.12–1.38
Never married		
Eligibility for basic livelihood right		
Yes	0.70	0.41–1.21
In the past	1.44	0.36–5.83
No	Referent	
Monthly income (KRW 10,000)		
<50	1.48	0.96–2.30
50–99	0.95	0.64–1.41
100–199	1.71	1.19–2.46
200–299	0.91	0.63–1.32
≥300	Referent	
Smoking type		
Smokers	1.06	0.65–1.73
Stop-smokers	0.75	0.51–1.11
Non-smokers	Referent	
Frequency of drinking alcohol (for the past year)		
Non-drinking	0.76	0.43–1.33
Less than once a month	0.66	0.39–1.11
Once a month	1.18	0.69–2.01
2–4 times a month	1.10	0.64–1.87
2–3 times a week	0.67	0.39–1.14
≥4 times a week	Referent	
Physical activities (non-performed/performed)	2.00	0.67–1.49
Body mass index (BMI)	1.02	0.98–1.07
Depression symptoms	1.09	1.04–1.15
Subjective health status		
Very bad	2.373	0.46–12.20
Bad	2.44	0.51–11.68
Neutral	2.98	0.63–14.09
Good	1.71	0.35–8.32
Very good	Referent	
Quality of life	0.90	0.82–0.99
Constant	0.01	

OR = odds ratio. CI = confidence interval. Bold numbers indicate significant values.

## Data Availability

Restrictions apply to the availability of these data. Data was obtained from the Korea Disease Control and Prevention Agency (KDCA) and are available from the author with the permission of the KDCA.

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
