# Peer review of "Analysis of Factors Associated with Subjective Mild Cognitive Impairment (MCI) among Older Adults Resident in the Community"

_ijerph, 2022, doi:10.3390/ijerph191610387_

Round 1

Reviewer 1 Report

.  This is a significant issue, as we have a problem of growing dementia not only in Korea but around the world, and any strategies that can impact mild cognitive impairment from occurring and possibly developing into dementia should be explored and given high priority.  Modifiable risk factors can be away to stave off this condition per these researchers and the data they have displayed.

Try to have a better lead and sentence in that regard to describe what you are talking about next, this is a minor grammatical issue.  My next, it is on line 333 when conclusions are being discussed.  It would be nice to have a summary sentence just including those factors that were found to be significant and they are impacting mild cognitive impairment and a sentence or 2, as a more detailed summary of what the researchers found.  Think it gives a better wrap to the article.

Author Response

Dear reviewer

Thank you for your review comments. I revised my manuscript according to your review. Please identify attached paper.

Sincerely 

From the researcher 

Reviewer 2 Report

The whole paper is written as if there were a casual relationship among MCI and the covariates. This is a cross-sectional non-experimental study that does not allow to make casual inference. So the last part of the Abstract (“The findings of this study show that MCI in older adults can be prevented…”) and the Conclusions section. .

There is need to improved the witing, avoiding reptition of verbs, phrases and words ingeneral.  (“it has been reported” lines 36 and 37; the term “data” lines 66 and 67).

In general the writing should be more simple with no so much details.

The statement “A term with the key word or meaning of “subjective” has been proposed for implementation to define between cognitive decline due to normal aging and due to MCI” with reference 4, is questinnable. There is not a general consensus about it. 

Section 2.3 can be reduced to a paragraph summarized in a flowchart and taking away such a long explanation about it. It is enough with: In this study, subjects with MCI were selected by applying Paterson's (correct the surname) diagnostic criteria of MCI (lines 88 and 89). 

Others sections to be reduced in extensión, avoiding redundancy are

2.2.2; 2.4.3; 2.4.4; 3.2 and the Discussion section. 

Author Response

(The authors gave the same response as above.)

Round 2

Reviewer 2 Report

I appreciate the modifications made which improve the quality of the article.